# Wheat Flour Pasta Combining *Bacillus coagulans* and *Arthrospira platensis* as a Novel Probiotic Food with Antioxidants

**DOI:** 10.3390/foods13213381

**Published:** 2024-10-24

**Authors:** Aldo Iván García-Moncayo, Emilio Ochoa-Reyes, Hilda Karina Sáenz-Hidalgo, Pedro González-Pérez, Laila N. Muñoz-Castellanos, David Roberto Sepúlveda-Ahumada, José Juan Buenrostro-Figueroa, Mónica Alvarado-González

**Affiliations:** 1Coordinación de Tecnología de Productos Hortofrutícolas y Lácteos, Centro de Investigación en Alimentación y Desarrollo, Cd. Delicias, Chihuahua 33089, Mexico; aldo94@live.com.mx (A.I.G.-M.); emilio.ochoa@ciad.mx (E.O.-R.); hsaenz@ciad.mx (H.K.S.-H.); jose.buenrostro@ciad.mx (J.J.B.-F.); 2Coordinación de Tecnología de Alimentos de la Zona Templada, Centro de Investigación en Alimentación y Desarrollo, A.C., Avenida Río Conchos s/n, Parque Industrial, Cd. Cuauhtémoc, Chihuahua C.P. 31570, Mexico; pedro.gonzalez@ciad.mx (P.G.-P.); dsepulveda@ciad.mx (D.R.S.-A.); 3Facultad de Ciencias Químicas, Universidad Autónoma de Chihuahua, Campus II Circuito Universitario s/n, Chihuahua 31125, Mexico

**Keywords:** wheat flour pasta, functional, *Arthrospira platensis*, probiotic, antioxidant, novel foods

## Abstract

*Arthrospira platensis* (Ap) and *Bacillus coagulans *(Bc) have been successfully used to develop functional foods, but a combination of both regarding functional implications in nutritional value and antioxidant capacity has not been explored. This work aimed to develop an artisanal wheat flour pasta with egg using 5% *A. platensis* and 1% *B. coagulans* GBI 6068 (labeled as Bc+Ap). Uncooked pasta was characterized regarding nutritional value; furthermore, total phenolic content, antioxidant capacity by 2,2′-azino-bis(3-ethylbenzothiazoline-6-sulfonic acid (ABTS), and ferric reducing antioxidant power (FRAP), pigment content, colorimetry assay, textural profile analysis, buffering capacity, and probiotic viability were carried out on uncooked and cooked pasta to assess the changes induced by cooking. The Bc+Ap pasta showed enhanced nutritional value with a significant increase in protein content (30.61%). After cooking, the pasta showed increased phenolic content (14.22% mg GAE/g) and antioxidant capacity (55.59% µmol Trolox equivalents/g and 10.88% µmol Fe^+2^/g) for ABTS and FRAP, respectively, as well as pigment content (6.72 and 1.17 mg/100 g) for chlorophyll a+b and total carotenoids, respectively, but relative impacts on colorimetric parameters in contrast to control (wheat flour pasta). Furthermore, Bc+Ap showed improved firmness (59%, measured in g), buffer capacity (87.80% μmol H^+^(g × ΔpH)^−1^), and good probiotic viability (7.2 ± 0.17 log CFU/g) after the cooking process.

## 1. Introduction

With increasing knowledge of how diet and diseases are related, the primary role of the diet has changed from supplying daily metabolic requirements to the use of the food itself as a way to promote health and reduce the risk of diseases [1]. However, although consumers are now more interested in nutrition and health, diet-related diseases, especially obesity, type 2 diabetes mellitus, hypertension, coronary heart disease, and metabolic syndrome, are still a concern [2]. Metabolic syndrome is characterized by impaired glucose tolerance (IGT) and hyperinsulinemia associated with high blood triglycerides (TG), high very-low-density lipoprotein (VLDL) levels, low high-density lipoprotein (HDL) concentrations, hypertension, and visceral adiposity, representing a dramatic public health concern whose final stage is often the development of non-communicable diseases (NCDs) if those deregulations are not treated at early stages [3].

NCDs are not contagious and often result from a complex interplay of genetic, environmental, and lifestyle factors, among them diet. In this context, the global patterns regarding NCDs show accentuated trends in deaths in the last two decades, especially towards obesity, followed by hyperlipidemia, type 2 diabetes mellitus, hypertension, and non-alcoholic fatty liver disease [4]. Due to its high prevalence, metabolic syndrome now represents a dramatic public health concern, and both the medical and scientific communities agree regarding the need to define strategies to stem this emerging pandemic [3].

Before such a scenario, a market niche focused on developing foods that could prevent diseases through bioactive substances from vegetables, fruits, and edible plants [5]; those foods are called functional. Functional foods are naturally occurring or processed foods with health-promoting or disease-preventing properties beyond their traditional nutritional value [6].

Microalgae are being studied as an innovative ingredient to produce functional foods due to their high nutritional value (vitamins, proteins, polysaccharides, minerals, enzymes, and fibers) and content in bioactive compounds such as peptides, polyphenols, essential amino acids, mono and polyunsaturated fatty acids, carotenoids, and phycocyanins [7], with properties as antioxidant, anti-inflammatory, antidiabetic, anticancer, and antimicrobial compounds [8]. Several studies have demonstrated that the inclusion of microalgae in food improved the nutritional value and the content of bioactive substances such as antioxidants [9,10,11], which are important for preventing oxidative damage upon cells caused by oxidative stress, a state in which oxidation exceeds the antioxidant systems in the body secondary to a loss of the balance between them, causing hazardous effects upon the body [12].

*Arthrospira platensis *(Ap) is a filamentous cyanobacterium of great biotechnological and nutritional importance that has been used successfully in developing food commodities, pharmaceuticals, and cosmetics [13]. Its inclusion in foods has shown increases in not only the nutritional value (especially proteins) but also its content in bioactive substances [9] such as antioxidants, water-soluble pigment-protein complexes (phycocyanin), carotenoids, phenolic compounds, and antioxidant enzymes (superoxide dismutase, catalase, and peroxidase) [14]. *A. platensis* possesses GRAS status (Generally Recognized as Safe) by the FDA and is intended for 0.5–3 g per serving in a wide range of food matrixes, including noodles and pasta [15]. Over the past years, there has been a significant and increasing trend in the use of *Arthrospira platensis* across several fields (food and feed, medicine, bioenergy, eco-fertilizer, etc.). This trend is particularly pronounced in the food industry, where a clear tendency of publications about its use in several food commodities has been evidenced by several studies, particularly for China, India, Brazil, and the United States, among others [16].

Pasta, a staple food of worldwide consumption, has an affordable price, popularity, palatability, and good nutritional characteristics. Hence, pasta represents a good vehicle for the inclusion of nutriments, a consideration also supported by the Food and Drugs Administration (FDA) and the World Health Organization (WHO) [17]. Previous studies dealing with the inclusion of *Arthrospira platensis* in pasta reported increased nutritional value (especially protein) [18], improved textural properties [19], and enhanced antioxidant capacity [20,21].

Probiotics are another ideal example of functional ingredients since, with their inclusion, many food products today are becoming functional [22]. This represents a way to develop new food probiotic commodities (non-just dairy) with proven beneficial properties regarding the characteristic dysregulations of metabolic syndrome, causing a reduction in the prevalence of hyperglycemia and a decreased tendency to develop hypertension, according to recent research, and therefore, probiotics are an important tool to stem the diseases associated with metabolic syndrome [3].

Usually, dairy products are the most popular vehicle of probiotics, including products such as yogurts, fermented milk, and cheese [23]. Milk fat enhances the viability of probiotics and their acid-bile tolerance [24]. Therefore, dairy foods are used as carriers of probiotics mainly due to their pH, buffering capacity, and fat content, which creates extra protection for probiotics when entering gastrointestinal tract and enhances their maintenance [25]. Buffering capacity and pH are significant factors influencing the survival of probiotics and their potential probiotic effects during gastric transit [26]. Buffering capacity refers the food’s ability to resist changes in pH. It is a remarkable aspect to consider in gastric digestion since it will impact the physiochemical breakdown of food [27].

There is an increasing interest in developing non-dairy probiotic foods for consumers with allergies like lactose intolerance or those who prefer low-cholesterol products. However, traditional and economic dynamics influence this context, sometimes making the employment of dairy products not feasible, making it necessary to supply them in other culturally compatible ways [28]. Several studies have demonstrated that some non-dairy products also possess buffer capacity, and they can be considered feasible food carriers, like cereal-based products, vegetables, meat, fish products, fruit and fruit juices, etc. [24].

The most common probiotic microorganisms used are *Bifidobacterium* and *Lactobacillus* [29], and several approaches have been considered to increase their viability [30]. However, these microorganisms do not always apply to alimentary matrixes requiring thermal processing or cooking. Spore-former lactic acid bacteria (SFLAB) have been presented as a way to produce probiotic food that can be heat-treated or cooked without losing its beneficial properties [31]. 

*Bacillus coagulans* GBI 30 6068 is a probiotic strain with multiple health benefits, such as reduction in gastrointestinal symptoms [32] and improvement of irritable bowel syndrome symptoms [33], and in combination with prebiotics (fermentable substrates), it has been found to cause improvements in dysbiosis [34], intestinal gas symptoms, and rheumatoid arthritis; improve the immune response to viral infections of the respiratory tract [35]; and enhance the digestion of proteins and carbohydrates like lactose and fructose [36]. 

Previous studies have successfully used *B. coagulans* GBI 30 6068 in the development of probiotic pasta formulations, showing that the microorganism is capable of withstanding pasta-making process conditions and cooking process and remain viable in enough quantity to be considered a probiotic food without affecting primary pasta properties, texture, or flavor [35,36,37]. However, there is no information regarding a pasta formulation with added *A. platensis* and *B. coagulans*. For that, the development of this pasta could be of interest since many studies have indicated that *A. platensis* can be used as a fermentable substrate [38,39], supporting the cell count and viability of microorganisms in probiotic foods [40,41] and increasing the content of bioactive substances with antioxidant properties and the overall nutritional value of foods Therefore, this study aimed to develop an artisanal soft wheat flour pasta with egg and with added *A. platensis* and *B. coagulans* and to evaluate the effects on the nutritional value, antioxidant capacity, pigment content, color, probiotic viability, buffering capacity, and textural properties of the pasta. 

## 2. Materials and Methods

### 2.1. Pasta Formulation and Nutritional Value

Pasta formulations were made by mixing standard white wheat flour (69.83%), water (18.16%), egg yolk (11.31%), and salt (0.70%) with (a) 1% of *B. coagulans* BC30 GBI 6068 (1 × 1013 CFU/g) freeze-dried (Bc pasta); (b) 5% of a commercially available product of *Arthrospira platensis* dried biomass (Ap pasta); and (c) both *B. coagulans* and *A. platensis* in the previously mentioned percentages (Bc+Ap pasta). A pasta formulation without *A. platensis* and without *B. coagulans* was used as control. The resultant doughs were kneaded for 10 min, rested for 90 min at room temperature (22 °C), and processed using an Italian pasta hand press of 15 cm (Vencort, model: 349003) into tagliatelle-shaped pasta strips with dimensions of 100 × 5 × 1 mm. Pasta strips were left to dry for 48 h at room temperature and used for the subsequent analyses on proteins (method 46-12.01; AOAC), fats (method 2003.06-2006; AOAC), carbohydrates (by difference), fiber (method 978.10; AOAC), moisture (method 948.12; AOAC), and ash (method 08-01.01; AOAC). Caloric value was calculated using the Atwater system by multiplying each macronutrient content by 4, 9, and 4 kcal for proteins, fats, and carbohydrates, respectively. Furthermore, portions of uncooked and cooked pasta (10 g boiled in 200 mL of distilled water for 8 min (time for “al dente”)) were freeze-dried for further analysis.

### 2.2. Preparation of Extracts for Antioxidant Capacity Assays

One gram of each freeze-dried paste was milled in a blender and added to assay tubes with 9 mL of methanol. Then, it was vortexed for 1 min, sonicated for 30 min at 30 °C (Sonicator VWR International, New York, NY, USA. model: 150D), and finally centrifuged (Eppendorf^®^, Hamburg-Nord, Germany, model 5804R) at 7563× *g* for 5 min at 20 °C. Aliquots of the resultant supernatants were transferred to microtubes and stored in dark and cold conditions (−80 °C) until their use.

### 2.3. Total Phenolic Content (TPC)

The TPC was determined using the rigorous methodology of López-Martínez et al. [42]. Briefly, 20 µL of the extract was mixed with 20 µL of Folin–Ciocalteu reagent (Sigma-Aldrich^®^, Livonia, MI, USA) and left to react for 5 min. After that, 20 µL of 0.01 M Na_2_CO_3_ was added and left to react for another 5 min, and finally, 125 µL of distilled water was added. The absorbance was measured at 790 nm using a microplate reader (Multiskan GO, Thermo Fisher Scientific, Vantaa, Finland). The TPC was calculated from linear regression using a gallic acid solution (0–1500 mg/L) and expressed as mg of gallic acid equivalents per gram of sample in dry weight (mg GAE/gdw).

### 2.4. Antioxidant Capacity

#### 2.4.1. ABTS (2,2-Azinobis-(3-ethylbenzothiazoline-6-sulfonate) Assay

The ABTS assay was carried out according to Buenrostro-Figueroa et al. [43]. Briefly, the ABTS free radical (ABTS^•+^) was obtained by mixing 5 mL of 7 mM ABTS with 2.5 mL of 2 mM K_2_S_2_O_8_ and subsequently incubated at room temperature for 12 h. To prepare ABTS^•+^ solution, the mixture was diluted with ethanol until an absorbance of 0.70 at 734 nm was obtained. The assay was performed in a 96-well microplate, in which 10 µL of the sample was mixed with 190 µL of the ABTS^•+^ and left to react for 1 min. Then, the absorbance was recorded at 734 nm using methanol as a blank in a microplate reader (Multiskan GO, Thermo Fisher Scientific, Vantaa, Finland). Antioxidant capacity was obtained by linear regression using a calibration curve of Trolox (0–800 µmol) and expressed as µmol of Trolox equivalents per gram of sample in dry weight (µmol TE/gdw).

#### 2.4.2. FRAP (Ferric Reducing Antioxidant Capacity) Assay

FRAP reagent was prepared by combining 25 mL of 0.3 M acetate buffer (pH 3.6), 2.5 mL of 10 mM TPTZ (Sigma-Aldrich^®^) in 40 mM HCl, and 2.5 mL of 20 mM ferric chloride. This mixture was incubated for 20 min at 37 °C before use. In a 96-well microplate, 6 µL of the extract was added with 18 µL of distilled water and 180 µL of FRAP reagent. The absorbance was measured at 593 nm (Multiskan GO, Thermo Fisher Scientific) using methanol as a blank, ensuring the accuracy of the results. The antioxidant power was determined by linear regression using Iron (II) sulphate heptahydrate solutions (0–3000 µmol) and expressed as µmol of Fe^+2^ per gram of dried weight (µmolFe^+2^/gdw) [44].

### 2.5. Pigments Determination

Adapting the methodology of Braniša et al. [45] to microplate, the content of chlorophyll a, chlorophyll b, and total carotenoids was determined. Briefly, 0.5 g of milled sample was added with 5 mL of acetone (100%) in assay tubes, followed by a sonication (Sonicator VWR model: 150D) of 3 min and subsequent centrifuging at 7563× *g* for 10 min (20 °C). The resultant supernatants were stored in amber microtubes at −80 °C until their use. In a 96-well microplate, 200 µL of each extract was added, and the absorbance was measured in a microplate reader (Multiskan GO, Thermo Fisher Scientific) at 662, 645, and 470 nm for chlorophyll a, chlorophyll b, and total carotenoids, respectively. Acetone (100%) was used as blank. For the estimation of each pigment content, the equations of [46] were used (Equations (1)–(3)), and the results are expressed as milligrams/100 g of dry weight (dw).
(1)$$Chlorophylla\left(µ\frac{g}{mL}\right)=11.75A662-2.35A645$$
(2)Chlorophyllb(µg/mL)=18.61A645−3.96A662
(3)$$Total\;Carotenoids\left(\frac{µg}{mL}\right)=\frac{1000A470-2.27\left(Chla\right)-81.4(Chlb)}{227}$$

### 2.6. Textural Profile Analysis (TPA) 

The TPA was performed according to Milde et al. [47] using a texturometer TA. XTplus (Stable Micro Systems Ltd., London, UK) and texture exponent lite version 4.0.130.0 (Stable Micro Systems Ltd.; 2007) software. The pasta was cooked as mentioned before (considering the AACCI Method 66-50.01, 2000). Three measurements were performed in each formulation. The TPA analysis was carried out in pasta strips of 4 cm in length, which were subjected to two compression cycles, using a 75 mm diameter flat-ended cylindrical probe. The test configuration was programmed to a speed of 0.5 mm/s, the pre-test speed was 1 mm/s, and the compression distance was 75% of the original size. From the force–distance curve of the pasta, we determined the firmness (g), adhesiveness (g × sec), springiness (%), gumminess (g), chewiness (g), cohesiveness (g), and resilience (%).

### 2.7. Buffer Capacity 

Buffer capacity was evaluated using the methodology of Mennah-Govela et al. [27]. For this, pasta (100 g) was cooked in 200 mL of distilled water for 8 min (enough for “al dente”), and then, the pasta was drained and pureed with a blender. Twenty grams of pureed pasta were placed in beakers of enough length to allow the entrance of the pH lector tip (Thermo Scientific, Waltham, MA, USA. ORION STAR A211). The initial pH was measured and recorded in three locations, and then, aliquots of 0.5–1 mL of HCl 0.16 M were added and mixed with the sample. After adding each HCl aliquot and mixing, the pH was measured again in three locations until reaching a final pH of 1.5. All pureed samples were measured in triplicate. 

### 2.8. B. coagulans GBI-30 Count

Ten grams of each pasta in 90 mL of phosphate buffer solution (PBS) were homogenized (Seward Stomacher 400, West Sussex, England) at 230 rpm for 30 s, and serial decimal dilutions were made. Each dilution was heat-treated in a water bath at 75 °C for 30 min. Finally, 1 mL of each dilution was spread-plated on GYE agar and incubated at 37 °C for 48–72 h. Typical *B. coagulans* colonies were counted and calculated as CFU/mL. 

### 2.9. Colorimetric Analysis of Pasta 

The color of raw and cooked samples was measured using a colorimeter (KONICA MINOLTA CM-600d Chiyoda, Tokio, Japón), and a commercially available semolina pasta was used as a reference. For this, 100 g of pasta was milled in a blender until it became a powder. The milled sample was placed in a white recipient, covering all recipient surfaces (a powder layer of 2 mm of thickness). The lecture tube of the colorimeter was placed above the powdered paste, and then the colorimeter emitted a beam of light. The yellowness index was calculated according to Belahcen 2022 [48].

### 2.10. Statistical Analysis 

All samples were analyzed in triplicate and expressed as a mean (*n* = 3) ± standard deviation. Statistics analyses were performed using SAS software (SAS 9.0). In case of nutritional value, a one-way analysis of variance (ANOVA) followed by a Tukey test (*p* < 0.05) was performed. Regarding antioxidant capacity assays, pigment content, color measurements, and TPA, raw and cooked pastas were analyzed separately. A one-way ANOVA followed by a Tukey test (*p* < 0.05) was used to measure the differences between the raw pasta in the previously mentioned parameters, and another one-way ANOVA followed by a Tukey test (*p* < 0.05) measured the differences between the cooked ones. 

## 3. Results and Discussion

### 3.1. Nutritional Value

Pasta represents an important food in human nutrition due to its high content of complex carbohydrates, low glycemic index, and some proteins. Previous reports dealing with the addition of *A. platensis* to pasta have found enhancements in the nutritional value directly proportional to the addition of the microalgae, although with high variability [18,19,21]. The nutritional values of pasta samples are shown in Table 1.

Significant changes between pasta formulations were observed in protein, carbohydrate, and fat content (*p* < 0.05). The protein content of all samples showed statistically significant increases in the order of Bc, Ap, and Bc+Ap, in contrast to the control. The Bc formulation showed an increase in protein content of 19.02%, followed by Ap pasta with 19.65%. Finally, the Bc+Ap pasta showed a higher protein increase of 30.61% compared to the control, indicating that both *A. platensis* and *B. coagulans* positively impacted the protein content.

It is well known that *A. platensis* has a high protein content, but more information about this needs to be provided on *B. coagulans*. Regarding the increase in proteins produced by adding *B. coagulans* found here, two previous studies dealt with adding *B. coagulans* in pasta [35,37]. However, these studies did not perform a proximal analysis on this parameter to determine if the addition of *B. coagulans* had any effect on the protein content. Nonetheless, other studies have stated that bacterial spores possess a coat mainly made of proteins, representing 50–80% of the total spore protein [49]. Thus, the protein increase evidenced here by the addition of *B. coagulans* is probably attributed to spore coat proteins, but more studies must be carried out to confirm this.

According to the FDA, the recommended daily value (%DV) of proteins based on a diet of 2000 kcal is 50 g, categorizing a food as high in proteins when it contains at least 20% of the %DV per serving. Bc+Ap pasta, in a portion of two ounces, contributes 16.35% to the daily recommended value of proteins. This pasta provides a significant portion of protein needs and offers additional benefits and the potential beneficial effects of *B. coagulans* on digestive and gut health. On the other hand, the carbohydrate content showed a statistically significant tendency to decrease, especially for Bc+Ap pasta. Since the carbohydrates were calculated by difference, the inclusion of other macronutrients (protein) by the adding of Bc+Ap together probably caused the decreased tendency showed in this parameter. Other parameters such as the fat content, caloric value, fiber, ash, and moisture did not show an effect attributable to adding *B. coagulans* or *A. platensis*.

Although other studies have shown that *A. platensis* produces increases in proteins, fats, carbohydrates, fiber, and ashes [19,21,50], in this study, just the protein parameter was increased statistically significantly. In contrast, the other parameters did not show a change related to adding the microalga. Regarding these findings, Lemes et al. [19] stated that the nutritional profile of *A. platensis* is highly dependent on factors such as the strain used, growth media, and freshness. Thus, this can explain the differences between the nutritional profile of pasta with added *A. platensis* reported in the literature and that of the formulations developed here.

### 3.2. TPC (Total Phenolic Content)

The TPC of the samples is shown in Table 2. All samples with added *B. coagulans* (Bc and Bc+Ap) presented significantly reduced TPC compared to the non-added *B. coagulans* samples, except cooked Bc pasta, which, although it presented lowered TPC, was not significantly different from the cooked control (*p* < 0.05). The Bc pasta showed a significant decrease in TPC values when uncooked of 23.28% in contrast to the control pasta. Interestingly, after the cooking process, Bc pasta recovered almost the same value of TPC as the control pasta, with no difference (*p* < 0.05) among them (43.50 and 41.95 mg GAE/g for control and Bc pasta, respectively).

The addition of *A. platensis* resulted in significant increases in the TPC. Uncooked and cooked Ap pasta showed increases of 9.64 and 28.34%, respectively, in contrast to uncooked and cooked control. Contrastingly to the uncooked control, the uncooked Bc+Ap pasta showed a decrease in TPC of 11.41% but a significative increase after cooking of 14.22% compared to the cooked control. On the other hand, the TPC of uncooked and cooked Bc+Ap pasta was 19.20 and 11% lower than uncooked and cooked Ap pasta (47.18 and 55.83 mg GAE/g, respectively). Nonetheless, when comparing both Bc+Ap and Ap cooked samples (as occurred with the control and Bc cooked pasta), the Bc+Ap pasta recovered some of its TPC values after cooking and remained near the values obtained by the AP pasta. However, it was significantly (*p* < 0.05) lower than the Ap sample.

Other studies dealing with adding *A. platensis* to pasta have reported increases in the TPC of 0.16–1.31 and 1.23 mg GAE/g for 5% *A. platensis* uncooked and cooked pasta, respectively, in contrast to controls [21,50]. The results presented here showed increased 4.15 and 12.33 mg/g for uncooked and cooked 5% *A. platensis* pasta. This later gain could be due to the increment tendency of the cooked pasta to rehydrate during the extraction of the bioactive compounds, releasing more of its internal components to the media in contrast to the uncooked pasta. Furthermore, as was discussed previously, the high susceptibility of *A. platensis* to cultivation variables, freshness, and the strain used can explain the high variability in the bioactive compound profile among the different commercially available *A. platensis* products, explaining the differences in the TPC and antioxidant capacity of the different products with added *A. platensis* reported in the literature. According to Park et al. [51], the variables involved during mass production, namely the drying process, food processing techniques, and storage conditions, can affect the content of pigments, TPC, and antioxidant capacity of *A. platensis* powders.

### 3.3. Antioxidant Capacity of Pasta

#### 3.3.1. ABTS (2,2-Azinobis-(3-ethylbenzothiazoline-6-sulfonate) Assay

The results of the ABTS assay are shown in Table 2. Adding of *A. platensis* resulted in a significant increase in the antioxidant capacity of samples with respect to the control and Bc pasta (*p* < 0.05). Uncooked pasta with added *A. platensis* showed an increased antioxidant capacity of 90.41% and 116.70% for Ap and Bc+Ap formulations in contrast to control and Bc pasta, respectively. Nonetheless, in the uncooked Bc+Ap sample, a synergic effect appeared, significantly increasing (*p* < 0.05) the antioxidant capacity of the sample by 11.70% in contrast to Ap pasta (17.69 µmol TE/g).

This finding suggests that *B. coagulans* can produce some organic acids or other antioxidant molecules using *A. platensis* as a fermentable substrate during the rest time of doughs since the Bc pasta did not show this effect. Although the exact temporal order for spore germination has not been determined in bacterial spores, the germination process (and the steps involved) is generally triggered by the presence of nutrients, amino acids, sugars, and nucleosides [52]. This could suggest an increased spore response to a media rich in amino acids, vitamins, minerals, antioxidants, and salts provided by adding *A. platensis* into pasta. Previous studies have evidenced that *A. platensis* can be used as a fermentable substrate for lactic acid bacteria, increasing the cell count, viability, and nutritional quality of fermented products [38]. This suggestion of organic molecules derived from the interaction between *A. platensis* and *B. coagulans* is also supported by the results of cooked samples. When comparing Ap and Bc+Ap cooked pasta, these organic compounds responsible for the increase in antioxidant capacity in the uncooked Bc+Ap sample were degraded, causing the antioxidant capacity of both samples to remain at similar levels of 14.01 and 13.63 µmol TE/g for Ap and Bc+Ap, respectively. However, both samples evidenced antioxidant capacity degradation after the cooking process, showing respective reductions of 20.80% and 31.02% for Ap and Bc+Ap samples but remaining 59.93% and 55.59% higher than the cooked control.

On the other hand, control and Bc pasta presented slight differences after the cooking process, with minimal changes in their respective antioxidant capacity in the uncooked treatment. A decrease of 5.70% was evidenced for control pasta and an increase of 1.42% for Bc pasta, but there was no resulting statistically significant change in antioxidant capacity (*p* < 0.05).

Similar results on the antioxidant capacity by ABTS of probiotic pasta made with *B. coagulans* were obtained by Fares et al. [35]. The authors compared the antioxidant capacity of three formulations: the first one consisting of durum wheat Vendetta hulled up to 4.5%; the second being one durum wheat Vendetta hulled up to 4.5% (78.5%), mixed and homogenized with 18% of enriched barley flour (11% b-glucan) and 3.5% of vital gluten powder; and the third one consisting of the previous formulation but with 1% of *B. coagulans* GBI-30, 6086 (10.3 log CFU/g). The authors found a significant (*p* < 0.05) increase of 33% in the antioxidant capacity of the probiotic pasta in contrast to the second. After cooking, the antioxidant capacity of the three samples remained at similar levels without significant changes. At this moment, this is the only study that has assessed the antioxidant capacity by ABTS (with a similar methodology as the one used here) of pasta with added *B. coagulans* after and before cooking, obtaining a similar synergic effect as that shown here. However, the authors did not attribute this increase to any synergic effect of the microorganism and the pasta components.

At this moment, there are no more previous studies assessing the antioxidant capacity of pasta with added *A. platensis* or with *B. coagulans* to compare for reference values regarding the findings presented here by ABTS assay. Hence, the information presented here could be helpful for further analyses of antioxidant capacity measurement by ABTS on similar formulations.

**Table 2 foods-13-03381-t002:** TPC, ABTS, and FRAP assay results of uncooked and cooked samples.

	Uncooked	Cooked	Method
Pasta Formulation (%)	(mgGAE/g)	(mgGAE/g)	TPC
Control	43.03 ± 0.32 ^b^	43.50 ± 0.68 ^c^	
Bc	33.01 ± 0.08 ^d^	41.95 ± 1.72 ^c^	
Ap	47.18 ± 0.05 ^a^	55.83 ± 0.92 ^a^	
Bc+Ap	38.12 ± 0.04 ^c^	49.69 ± 0.95 ^b^	
	(µmolTE/g)	(µmolTE/g)	ABTS
Control	9.29 ± 0.10 ^c^	8.76 ± 0.08 ^c^	
Bc	9.13 ± 0.29 ^c^	9.26 ± 0.13 ^b^	
Ap	17.69 ± 0.61 ^b^	14.01 ± 0.09 ^a^	
Bc+Ap	19.76 ± 0.19 ^a^	13.63± 0.07 ^a^	
	(µmolFe^+2^/g)	(µmolFe^+2^/g)	FRAP
Control	13.67 ± 0.39 ^c^	11.3 ± 0.29 ^c^	
Bc	10.20 ± 0.55 ^d^	5.71 ± 0.12 ^d^	
Ap	21.61 ± 1 ^a^	14.76 ± 0.10 ^a^	
Bc+Ap	17.07 ± 0.74 ^b^	12.53 ± 0.05 ^b^	

Values are means (*n* = 3) ± SD. Different letters represent a statistically significant difference among treatment (*p* < 0.05). Bc = *Bacillus coagulans* pasta; Ap = *A. platensis* pasta; Bc+Ap = pasta combining Bc+Ap; TPC = total phenolic content; ABTS = 2,2′-azino-bis(3-ethylbenzothiazoline-6-sulfonic acid assay; FRAP = ferric reducing antioxidant power assay.

#### 3.3.2. Ferric Reducing Antioxidant Power (FRAP)

FRAP assay results are shown in Table 2. Contrary to the ABTS assay, adding *B. coagulans* caused a significant reduction (*p* < 0.05) in the ferric reducing antioxidant power capacity of uncooked and cooked samples in contrast to pasta without the microorganism. Uncooked Bc pasta showed a significantly lower ferric reducing antioxidant power value of 25.38% less than control pasta (13.67 µmol Fe^+2^/g). After cooking, although both the control and Bc pasta lost some of their original ferric reducing antioxidant power (17.33% and 44.01%, respectively), the Bc pasta obtained the lowest value (5.71 µmol Fe^+2^/g).

Regarding pasta with added *A. platensis* and *B. coagulans*, a similar effect was caused by *B. coagulans * addition. In this case, the synergic effect between *A. platensis* and *B. coagulans* observed previously in the ABTS assay was not shown here. The Bc+Ap sample showed lowered ferric reducing antioxidant power in uncooked (21%) and cooked (15.10%) in contrast to Ap pasta (21.61 and 14.77 µmol Fe^+2^/g for uncooked and cooked Ap pasta, respectively). However, both Ap and Bc+Ap samples obtained higher values of ferric reducing antioxidant power capacity in uncooked (58.08 and 67.35%) and cooked (30.70 and 119.43%) in contrast to control and Bc pasta, respectively, whose obtained values were 13.67 and 10.20 µmol Fe^+2^/g for uncooked and 11.30 and 5.71 µmol Fe^+2^/g for cooked control and Bc pasta, respectively.

Previous studies assessing FRAP values on *A. platensis*-added pasta have varied to a great extent, with values from 2.52 to 11.3 µmol/g and 2.96 to 31.4 µmol/g for uncooked control and 5% *A. platensis*-added pasta, respectively [21,50]. Regarding cooked pasta, values of 9.45 and 26.81 µmol/g for control and 5% *A. platensis* were reported [21]. Therefore, the results presented here are in the range of the FRAP values reported in the literature.

### 3.4. Spectrophotometric Estimation of Pigment Content

*Arthrospira platensis* is a natural source of pigments with recognized health benefits and antioxidant compounds such as chlorophylls, carotenoids, and phycocyanin (blue pigment), to which the strong antioxidant properties of *A. platensis* are attributed [53]. The pigment content provided by adding *A. platensis* to pasta is shown in Table 3.

The addition of *A. platensis* led to a unique enrichment of the pasta with pigments such as chlorophylls and carotenoids. The uncooked Ap pasta demonstrated a content of 5.16 and 1.5 mg of chlorophyll a+b and total carotenoids, respectively. However, the cooked Ap pasta exhibited a significant increase in all pigment content, with a 117.44 and 36.66% rise for chlorophyll a+b and total carotenoids, respectively.

The increase in pigment content after the cooking is probably attributed to the structural damage on *A. platensis* induced by the cooking process, releasing more of its internal components to the media. In addition, the cooking process produced a weakened gluten network in the pasta by the swelling of starch granules [54], producing a brittle pasta more permeable to the extraction media than uncooked pasta. Experimentally, cooked pasta was prone to rehydrate more easily and faster than uncooked samples, resulting in an increased release of the bioactive compounds contained in them to the extraction media.

Surprisingly, the addition of *B. coagulans* to pasta (Bc+Ap) caused a significant modification in the content of all pigments. In contrast to uncooked Ap pasta, the uncooked Bc+Ap pasta showed significant initial reductions of 15.70% and 8.66% for chlorophyll a+b and total carotenoids, respectively. Furthermore, when comparing cooked Ap pasta with cooked Bc+Ap pasta, significant reductions (*p* < 0.05) of 40.10 and 42.92% were also noticed for chlorophyll a+b and total carotenoids, respectively. On the other hand, the increases in chlorophyll a+b and total carotenoids evidenced by Ap pasta after cooking were different for Bc+Ap pasta. Instead, an increase of 54.48% and a reduction of 14.60% were found for chlorophyll a+b and total carotenoids, respectively, in Bc+Ap pasta.

Although there are just a few studies assessing the pigment content regarding chlorophylls and total carotenoids in foods with added *A. platensis* [55,56,57] to compare for reference values, the results and addition percentages assessed here are similar only to those reported by Tańska et al. [56], who developed corn extrudates with several percentages of *A. platensis* by using an extrusion-cooking process. The authors found chlorophyll values of 6.2 and 12.6 mg/100g for corn extrudates added at 4 and 6%, respectively. Therefore, the value of 11.22 mg/100g for chlorophyll obtained by cooked pasta added at 5% fits between the range of the values reported by these authors. Nonetheless, regarding carotenoid content, there are no reference values in similar percentages to the addition of *A. platensis* assessed here.

### 3.5. Textural Profile Analysis of Pasta

The result of the TPA is shown in Table 4.

The hardness parameter increased by adding *A. platensis* and *B. coagulans* and the combination of both, in contrast to the control in raw and cooked pasta. For raw pasta, the most significant increase in hardness was shown by the addition of *B. coagulans *(72.69%), followed by Ap pasta (32.03%), and, finally, the combination of both in pasta Bc+Ap (17.15%). Cooked pasta tended to increase in hardness compared to raw pasta, with the highest value obtained by the Bc+Ap sample (59.32%). These results suggest a superior improvement in the cooked pasta structure compared to all other cooked formulations. According to Ogawa and Adachi [58], hardness is a parameter governed by the strength of the gluten network. Therefore, combining *A. platensis * and *B. coagulans* improved the gluten network structure, enhancing its strength. 

Chewiness (product of hardness, cohesiveness, and springiness) and gumminess (product of hardness and cohesiveness) are related parameters referred to as the necessary energy to be applied to disintegrate a pasta fragment and subsequently swallow it. In this case, these parameters were also significantly affected by adding *A. platensis* and *B. coagulans*. In raw pasta, the highest chewiness value was obtained by Ap pasta, reaching an increase of 32.71%, followed by Bc pasta with 3.58%, and, finally, Bc+Ap pasta with a reduced chewiness value of 36.57% in contrast to the control. However, all cooked pasta was enhanced in chewiness compared to the control, reaching values of 31.63%, 58.93, and 57.25% for Bc, Ap, and Bc+Ap, respectively. 

Regarding gumminess, raw pasta showed increases of 57.57, 30.88, and 24.68% for Bc, Ap, and Bc+Ap, respectively, in contrast to the control. In cooked pasta, the highest value in gumminess was obtained by Ap pasta at 59.07%, followed by Bc+Ap at 47.29%, and Bc at 19.52%. 

In general, pasta must meet the consumer’s requirements, meaning that the product must retain its color, have a smooth surface, and be firm and elastic [59]. All cooked pasta developed here showed increased firmness (hardness), chewiness, and gumminess compared to control pasta. The main factor behind the improvement of hardness and thus its related parameters is the increase in proteins attributable to the addition of *A. platensis*. According to Sozer et al. [60], firmness and adhesiveness are the most critical textural parameters in cooked pasta quality. The protein fraction, in particular, mainly influences the firmness of pasta and tolerance to overcooking. The authors found that spaghetti with reduced protein content absorbed more water, resulting in high stickiness and low firmness. Therefore, the enhancement of the hardness of pasta is due to the contribution of proteins from *A. platensis* to the pasta gluten network. Zouari et al. [20] obtained similar results and found increased firmness in cooked semolina pasta added with 2% *A. platensis*. The authors attributed this enhancement to the embedding of gelatinizing starch granules in a gluten network with the coexistence of microalgal proteins provided by this cyanobacterium. This effect can be explained by Nilusha et al. [17], who stated that a gluten network is formed when glutenin and gliadin are exposed to water, and when pasta is cooked, two events determining pasta properties can occur. According to the authors, a uniform and compact gluten network with swelled starch granules is formed when pasta is cooked. However, a physical competition between the coagulation of proteins and starch swelling occurs. If the protein coagulations are predominant during cooking time, then the starch granules are trapped in an alveoli-like system, enhancing the firmness of pasta. The contrary occurs when starch hydration wins, resulting in pasta with non-abrasiveness and typical stickiness. 

### 3.6. Buffer Capacity of Pasta

The buffering capacity is an important parameter for increasing the survival of probiotics. Typically, dairy products (mainly the fermented ones) are used as vehicles (also known as carriers) to carry probiotics due to characteristics and requirements such as low-temperature needs (4 °C–8 °C), limited shelf life (15–25 days), their richness in the necessary nutrients to promote the growth of the probiotic microorganisms, and the easy availability of the necessary guidelines for the use of probiotics in dairy products [61]. However, the main aspect that makes dairy products the preferred vehicle for the delivery of probiotics is their buffer capacity [25], which is attributed to their content of protein and minerals (calcium, citrate, phosphate, and lactate) [62].

Buffering capacity refers to the ability of a substance to resist changes in pH [27]. The relationship between buffer capacity and probiotic viability relies on the capacity of the food to slow down the velocity at which food changes its pH. Hence, the greater the buffer capacity, the more gastric acid concentration and time needed to change the pH of the food. This delay in the time required to change the pH can be visualized as a less drastic change in the pH; this means that the changes in the food are more gradual and, therefore, less stressful and lethal for probiotic microorganisms. Buffering capacity is essential from the point of view of digestion and storage/shelf life. 

Since the gastrointestinal passage is a harsh environment characterized by high gastric acidity, oxygen stress is induced by ROS (Reactive Oxygen Species) released from mucosal surfaces, bile salt stress, and osmotic stress, among other factors. The nature of the food when probiotics are added becomes a vital factor regulating its further colonization in the gastrointestinal tract and aiding probiotic bacteria by buffering the stomach’s acidic environment and supporting its viability alone or by adding other functional ingredients that can improve this capacity [63]. 

The study of Mennah-Govela et al. [62] indicated that the buffering capacity depends on the food’s composition; this means protein content and content of amino acids such as aspartic and glutamic acid, organic acids, initial pH, and other food ingredients, additives, as well as food particle size (an increased superficial area also increases buffer capacity). Furthermore, the authors indicated that fat and carbohydrates negatively affect buffer capacity, preventing acid reactions with proteins by interfering with the diffusion of H^+^ or impacting reactions of H^+^ with those compounds with inherent buffering capacity. 

The acid-titration curves and buffering capacity of the pasta as well as the macronutrients that affect the buffering capacity obtained from the nutritional analysis are shown in Figure 1 and Table 5, respectively. All samples had similar pH values at the beginning (6.27, 6.10, 6.30, and 6.28 for control, Bc, Ap, and Bc+Ap samples, respectively). However, the behavior during acid titration was different. The lowest buffer capacity was shown by control pasta, followed by Bc, Ap, and, finally, the Bc+Ap formulation, which obtained the highest value of the assay. Adding *B. coagulans *(Bc pasta) resulted in an increased buffer capacity of 3.30% in contrast to control pasta. In comparison, adding *A. platensis* (Ap pasta) yielded an increase of 11.30%. Surprisingly, the combination of both *A. platensis* and *B. coagulans* resulted in an enhanced buffer capacity compared to *A. platensis* or *B. coagulans* alone, reaching a value of 87.84% in contrast to the control.

The differences between the formulations can be explained by their composition according to the information previously discussed. The control pasta obtained the lowest buffering capacity, attributable to the reduced protein content and increased fat and carbohydrate content. On the other hand, Bc pasta increased its buffer capacity by 3.3% in contrast to the control, probably attributable to its increase in proteins (by the nitrogen introduced by the microorganism addition) and reduced fat and carbohydrate content. 

A slight increase in the buffer capacity was shown in the Ap pasta, clearly attributable to the enrichment of the pasta with the protein content *A. platensis*. According to the nutritional analysis carried out by several studies, glutamic and aspartic acid are two of the most abundant amino acids found in this cyanobacterium [64,65], information which is also in agreement with the nutritional value of the product used for the development of the pasta in this work. 

On the other hand, when *A. platensis* and *B. coagulans* were mixed into the pasta (Bc+Ap), the buffering capacity rose higher than the values obtained by *A. platensis* or *B. coagulans* alone, suggesting a kind of synergic effect.

Although the study of Mennah-Govela et al. [27] showed the importance of a low content of fat for improving buffering capacity in gastric conditions, the study of Tompkins et al. [66] gave a broad assessment of the impact of fat content on the viability of probiotic microorganisms. Between their experiments, the authors assessed in vitro the effect of the buffering capacity of the food on the survival of probiotic bacteria (*Lactobacillus helveticus* R0052, *Lactobacillus rhamnosus* R0011, *Bifidobacterium longum* R0175, and *Saccharomyces cerevisiae boulardii*) during gastrointestinal transit. They found that even small percentages of fat (1% *w*/*w*) greatly impacted the number of viable bacteria reaching the duodenum, suggesting a protective effect by fat and proteins against bile and pancreatic enzymes. The authors indicated that regarding the viability of probiotics, fat was most important, followed by proteins. However, the carrier must provide both in a correct ratio to ensure probiotic viability. 

The high resistance of *B. coagulans* to harsh gastric conditions has been recognized by several studies and has been defined to vary in a strain-dependent way. Nonetheless, the study of Majeed et al. [67] assessed the survival of spores of *B. coagulans* MTCC 5856 to gastric acid, and although they did not find significant differences in the spore count at a pH of 3–8, in contrast to the initial spore count (10 log^10^ spores/g) up to 4 h, a decrease of 0.9 and 2.1 log 10 reduction was evidenced at a pH of 1.5 in 1 and 4 h, respectively. However, there is information about the effect of gastric conditions at several acidic pH levels for *B. coagulans* GBI 30 6068 cells, showing reductions between 16% to 33.33% at pH values of 3 to 1, respectively, from an original count of 6 log CFU/mL [37]. At this moment, there is no similar information regarding the gastric acid resistance of *B. coagulans* GBI 30 6068 spores at several acidic pH values, as was researched in the study of [67]. However, since the Bc+Ap pasta obtained good buffering capacity, the gradient of pH change should be lesser, and therefore, enhanced viability may be expected, but further analysis in vitro mimicking full gastrointestinal conditions is needed to assess this effect and the impact on *B. coagulans* viability. This finding could be relevant as a way to maintain the integrity of most of these spores along the harsh conditions of the digestion process to increase the number of intact spores reaching the duodenum, defined as the active site where the spore germination occurs [29].

### 3.7. B. coagulans: Probiotic Viability

Probiotics are defined as “live microorganisms that, when administered in adequate amounts, confer a health benefit on the host”. However, to exert their beneficial properties, probiotics have to be administered at least at the “minimum therapeutic” level, reached when supplied at 10^6^ CFU/g of viable cells [68]. While *Lactobacillus* and *Bifibacterium* species have been typically used as probiotics and have shown remarkable probiotic activities, their survival is commonly low, ranging from 1–15% or even lower in some strains [29]. These low survival rates pose significant challenges to the viability of probiotics, especially considering the harsh conditions they undergo from manufacturing methods, storage, and shipping conditions as well as the normal physiological conditions (acidic environment of stomach and bile salts when they are consumed); these are some aspects responsible for their low viability [31]. However, probiotic spore-forming bacteria, with their ability to withstand the effects of food and feed processing [69], offer a promising solution. Their spore-former ability brings them the capacity to withstand harsh conditions in contrast to vegetative cells [70]. The results of the viability assay of the pasta with added *B. coagulans*, a probiotic spore-forming bacteria, are shown in Table 6.

The viability of *B. coagulans* behaved differently in the presence of *A. platensis*. Although the initial concentration of added *B. coagulans* was 13 log CFU/g, the number of colonies on agar plates showed for Bc pasta a count of 5.19 log CFU/g on average, whereas when *B. coagulans* was in combination with *A. platensis*, an increased response was evidenced, showing a count of 7 log CFU/g. Reductions of 7.81 and 6 log were evidenced by comparing the initial concentration of *B. coagulans* in the freeze-dried (13 log CFU/g) and the final concentration of raw pasta by the results of the agar plating for Bc and Bc+Ap pasta, respectively, without significative changes in the *B. coagulans* counts after the cooking process (8 min).

The differences between the counts of *B. coagulans* for Bc and Bc+Ap pasta is probably attributed to the compounds contained in *A. platensis*. According to Løvdal et al. [71], dormant bacterial spores of bacillus can be induced to germinate by nutrients such as amino acids, purine nucleosides, sugars, ions, and combination of these. Hence, although Bc+Ap pasta had a higher count of CFU/g on the agar plates in contrast to Bc pasta, both formulations contained the same amount of CFU/g, but more spores were triggered to germinate by the presence of the previously mentioned compounds that are also present in *A. platensis* in contrast to Bc pasta. 

Previous studies on probiotic pasta with *B. coagulans* have shown the high viability of this microorganism to withstand the making and cooking process and remain viable in enough quantities to be considered a probiotic food (>1 × 10^6^). Fares et al. [35] developed probiotic pasta with 1% of freeze-dried *B. coagulans *(10 log CFU/g). The authors obtained a concentration of 8 log CFU/g when *B. coagulans* was mixed with wheat flour and a final concentration of 7 log CFU/g in raw pasta. Furthermore, the authors assessed the effect of cooking on the counts of *B. coagulans* colonies. They found reductions of 0.43 and 0.62 log CFU/g after cooking for 5 and 7 min, respectively, in contrast to raw pasta (7.34 log CFU/g). A deeper analysis of the changes in *B. coagulans* CFU counts during the pasta-making process was described in the study of Konuray and Erginkaya [37]. The authors added *B. coagulans* to wheat flour until they obtained an approximately 8.62 log CFU/g concentration. Subsequently, they measured the concentration after the formation of the dough, extrusion, drying, and even after six months of storage, finding minimal reductions of 0.12, 0.01, 0.19, and 0.96 log CFU/g, respectively. These findings evidence the high viability of this microorganism in producing probiotic pasta products without detrimental effects on its CFU count even after 6 months of storage. 

The results of this study, as shown by Bc+Ap pasta regarding the bacterial count and viability after the cooking process, are in agreement with those of the previously mentioned authors, who obtained counts of 8–7 log CFU/g for raw pasta and 7 log CFU/g for cooked pasta. These counts represent enough quantities of CFU per gram to confer the potential health benefits associated with probiotic foods. 

### 3.8. Colorimetric Analysis

The colorimetric analysis of pasta is shown in Table 7.

The use of Ap in pasta formulations caused an enrichment of mainly dark-green pigments such as chlorophyll and others such as blue pigments (phycocyanins) and yellow–orange pigments (carotenoids and lycopene), among others. Pasta with added Ap showed significant changes in L* (lightness), a* (− red to green +), and b* (− yellow to blue +) values attributable to their content in pigments. On the other hand, adding *B. coagulans* to pasta did not cause visually appreciable changes in pasta color compared to the control and commercial control.

After cooking, pasta with added Ap showed increased redness in color, probably due to the content of yellow–red pigments such as carotenoids and lycopene. Those pigments are more thermally stable in contrast to chlorophyll, suggesting a change in the chlorophyll color dominancy after cooking. Furthermore, the yellow index (an indicator of yellowness) was highly increased in pasta after cooking, which could be attributable to the prevalence of yellow pigments such as carotenoids. The changes induced by Ap addition, therefore, resulted in relative changes in the color of pasta, bringing them an attractive green color.

### 3.9. Effect of B. coagulans Addition on Total Phenolic Content, Antioxidant Capacity, and Pigment Content of Pasta

Regarding TPC and antioxidant capacity assays, adding *B. coagulans* caused significant reductions in the TPC and FRAP values of uncooked and cooked pasta. However, a synergism in the ABTS assay was evidenced in the Bc+Ap pasta. Uncooked Bc+Ap pasta showed 19.20% less TPC and 21% less FRAP values but a significative increase of 11.70% in the ABTS (compared to uncooked Ap pasta). After cooking, Bc+Ap pasta showed 11 and 15.10% lesser TPC and FRAP values, respectively, but just a 2.71% difference regarding the ABTS assay in contrast to cooked Ap pasta. Nonetheless, cooked Bc+Ap pasta showed 14.22, 10.88, and 55.60% more TPC, FRAP, and ABTS values than cooked control pasta.

This significative pattern of reduction in the TPC and FRAP but not significant for ABTS induced by *B. coagulans* could be due to interference produced by the bacterial spores. Currently, no previous reports have assessed the effects caused by *B. coagulans* on the antioxidant capacity of pasta samples with added *A. platensis* by ABTS, FRAP, and TPC, nor have any assessed the impact on pigment content in similar foods. However, some previous reports assessing the chemical properties of bacterial spores can be valuable given the need for more information to explain the phenomenon presented here, and this has also not been described elsewhere in other studies at this moment.

The study of Gerhardt and Black [72] using dormant spores of *Bacillus cereus* strain terminalis demonstrated that bacterial spores could uptake chemical compounds from media based on their charge, molecular weight, and lipophilicity. No physiological specificity towards those compounds of importance to the spore (metabolizable compounds or those needed for germination) was evidenced.

Although there is no information regarding this effect on *B. coagulans* spores, if this is the case, since *A. platensis* is highly dense in compounds such as proteins, amino acids, antioxidants, vitamins, minerals, and pigments, it is not easy to assess which one of them is being uptake by the spores and the mechanisms involved. Possibly, molecules of low molecular weight and lipophilicity should be targeted as the preferred candidates, as indicated by Gerhardt and Black [72]. Furthermore, it is considered that the charge of the molecule is important.

Regarding the charge of the compounds, Douglas [73] suggested that dormant spores possess a surface covered by amino and carboxyl groups. The author suggested that at a low pH, these groups exist as -COOH and -NH3+, at neutral pH as -COO- and -NH3+, and at basic pH as -COO- and -NH2. Hence, it is suggested that at low pH, the spore surface tends to ionize, producing a positive net charge; at neutral pH, the spore surface behaves as a zwitterion (net charge equals 0) with affinity to adsorb anions; and the contrary effect of low pH should occur in basic conditions. Hence, depending on the pH media conditions, different molecules can be targeted to be uptaken by the bacterial spores.

Nonetheless, in the antioxidant capacity assays, the oxidizing/reductant agent must be free to interact with the antioxidants of *A. platensis*. Since each antioxidant capacity assay was carried out at a different pH, the bacterial spore probably produced interferences depending on the pH. As was discussed previously, the effect induced by the superficial charges of the spore by amino and carboxyl groups could be having a considerable impact during the performing of the TPC and antioxidant capacity assays, probably affecting the interaction between the oxidizing or reductant agents with the antioxidants of *A. platensis* since the spore coat is mainly constituted by proteins (representing the 50–80% of the total protein content of the spore) [49,74].

It is out of the scope of this work to determine the interaction between *B. coagulans* spores and the TPC and antioxidant capacity assays. However, based on the previously discussed studies, it is suggested that the phenomenon responsible for the modifications to the antioxidant capacity of Bc pasta is probably induced by the resultant charges on the spore surface depending on the pH of the media from the antioxidant capacity assay (i.e., interfering with the electron donor or acceptor capacity between the oxidant or reductant agent and the antioxidants contained in *A. platensis*).

All pigments showed a reduction in their content when *B. coagulans* was present in pasta but to different extents. This effect on the reduction in the pigments could be explained considering the statement by Gerhardt and Black [72] that the molecular weight and lipophilicity of pigments could show affinity to spores. Nonetheless, more studies assessing this interference/absorption phenomenon are needed to understand the fundamental nature of the process involved.

This is the first work assessing the changes induced by a probiotic microorganism regarding nutritional value, TPC, antioxidant capacity, pigment content, TPA, buffering capacity, probiotic viability, and colorimetry. Therefore, the methodologies, information, and findings presented here could be helpful for further analysis of similar probiotic foods as a reference guide to measuring the impacts of adding spore-former probiotic microorganisms and the changes induced not only from the nutritional point of view but also from the textural and functional properties.

## 4. Conclusions

The combination of *A. platensis* and *B. coagulans* in pasta produced a food of increased nutritional value regarding proteins (30.61%), enriched with functional pigments such as chlorophylls and carotenoids (6.72 and 1.17 mg/100 g), and with probiotic potential since it supplies enough viable *B. coagulans* cells in cooked pasta to be considered a probiotic food (7 log CFU/g). In addition, the buffering capacity assay demonstrated that this formulation (Bc+Ap) exhibited the highest buffering capacity (87.80% in contrast to control), which could help maintain the integrity of most of the spores of *B. coagulans* during their transit through harsh gastric conditions. Nonetheless, further analyses in vitro mimicking full gastrointestinal conditions are needed to assess this effect and its impact on *B. coagulans* viability. On the other hand, cooked Bc+Ap pasta showed the best firmness (59.33%), chewiness (57.24%), and gumminess (47.28%), thus obtaining a nutritionally improved food with probiotic potential and enhanced textural properties and antioxidants.

Furthermore, studies assessing the benefits of this formulation regarding the parameters of metabolic syndrome could be of interest to measure the impact of introducing this formulation on the dietary patterns of subject studies on the characteristic early stages of dysregulations that are conducive to the development of non-communicable diseases.

## Figures and Tables

**Figure 1 foods-13-03381-f001:**
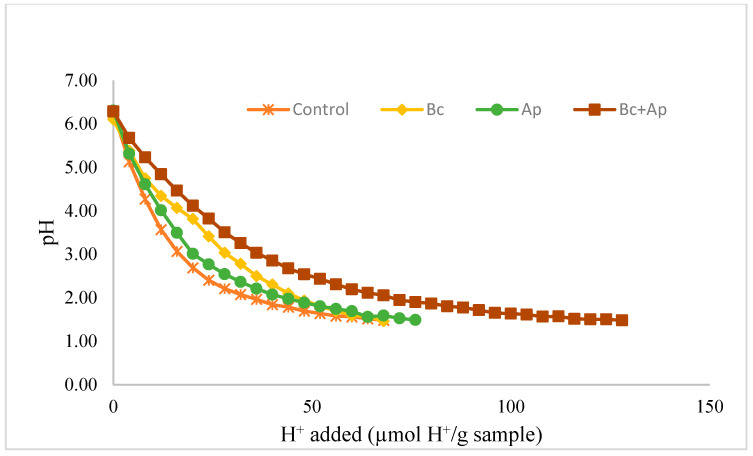
Acid-titration curves of cooked pasta. Each dot represents a pH measurement performed in triplicate.

**Table 1 foods-13-03381-t001:** Proximal analysis per 100 g of uncooked sample (dw).

Component/Pasta	Control	Bc	Ap	Bc+Ap
Moisture (%)	11.55 ± 0.05 ^a^	11.68 ± 0.15 ^a^	11.64 ± 0.15 ^a^	11.49 ± 0.09 ^a^
Ash (g)	1.39 ± 0.02 ^a^	1.45 ± 0.03 ^a^	1.53 ± 0.05 ^a^	1.33 ± 0.18 ^a^
Crude Fiber (g)	2.84 ± 0.21 ^a^	2.80 ± 0.23 ^a^	2.65 ± 0.14 ^a^	2.55 ± 0.24 ^a^
Fat (g)	3.76 ± 0.27 ^a^	3.08 ± 0.09 ^b^	3.39 ± 0.29 ^ba^	3.41 ± 0.11 ^ba^
Protein (g)	11.04 ± 0.63 ^b^	13.14 ± 0.62 ^a^	13.21 ± 0.43 ^a^	14.42 ± 0.59 ^a^
Assimilable carbs (g)	69.42 ± 0.73 ^a^	67.84 ± 0.99 ^ba^	67.58 ± 0.82 ^ba^	66.79 ± 0.57 ^b^
Energy (Kcal)	355.69 ± 0.77 ^a^	351.64 ± 0.87 ^c^	353.7 ± 0.76 ^b^	355.56 ± 0.59 ^ba^

Values are means (*n* = 3) ± SD. Different letters within the same column represent a statistically significant difference (*p* < 0.05). Bc = *Bacillus coagulans* pasta; Ap = *A. platensis* pasta; Bc+Ap = pasta combining Bc+Ap; dw = dry weight.

**Table 3 foods-13-03381-t003:** Chlorophyll a + b and total carotenoids content of pasta samples (mg per 100 g of dw).

Pasta Formulation	Uncooked	Cooked
	Chlorophyll a + b (mg /100 g)	
Ap	5.16 ± 0.07 ^a^	11.22 ± 0.06 ^a^
Bc+Ap	4.35 ± 0.03 ^b^	6.72 ± 0.07 ^b^
	Total carotenoids (mg /100 g)	
Ap	1.5 ± 0.01 ^a^	2.05 ± 0.03 ^a^
Bc+Ap	1.37 ± 0.04 ^b^	1.17 ± 0.09 ^b^

Values are means (*n* = 3) ± SD. Different letters represent a statistically significant difference among treatment. (*p* < 0.05). Bc = *Bacillus coagulans* pasta; Ap = *A. platensis* pasta; Bc+Ap = pasta combining Bc+Ap.

**Table 4 foods-13-03381-t004:** Textural Profile Analysis of cooked pastas.

Pasta Formulation	Hardness (g)	Adhesiveness (g.sec)	Springiness (%)	Cohesiveness (g)	Gumminess (g)	Chewiness (g)	Resilience (%)
Control	5262 ± 190 ^d^	−147 ± 19 ^a^	0.75 ± 0.04 ^a^	0.81 ± 0.03 ^ba^	4278 ± 155 ^c^	3197 ± 286 ^b^	0.2 ± 0.05 ^a^
Bc	6077± 256 ^c^	−283.3 ± 74 ^b^	0.83 ± 0.1 ^a^	0.84 ± 0.1 ^ba^	5113 ± 630 ^bc^	4208 ±146 ^ba^	0.19 ± 0.03 ^a^
Ap	7398 ± 177 ^b^	−132 ± 9 ^a^	0.74 ± 0.05 ^a^	0.92 ± 0.07 ^a^	6805 ± 444 ^a^	5081± 542 ^a^	0.223 ± 0.04 ^a^
Bc+Ap	8384 ± 173 ^a^	−175 ± 5 ^ba^	0.81 ± 0.19 ^a^	0.75 ± 0.06 ^b^	6301± 480 ^ba^	5027 ± 882 ^a^	0.21 ± 0.1 ^a^

Values are means (*n* = 3) ± SD. Different letters represent a statistically significant difference among treatment (*p* < 0.05). Bc = *Bacillus coagulans* pasta; Ap = *A. platensis* pasta; Bc+Ap = pasta combining Bc+Ap.

**Table 5 foods-13-03381-t005:** The total buffering capacity of pastas and macronutrients involved.

Pasta Formulation	Total Buffering Capacity μmol H+ (g × ΔpH)^−1^	Protein Content (g/100 g)	Fat Content (g/100 g)	Carbs Content (g/100 g)
Control	14.19 ± 0.12 ^d^	11.04 ± 0.63 ^b^	3.76 ± 0.27 ^a^	69.42 ± 0.73 ^a^
Bc	14.66 ± 0.03 ^c^	13.14 ± 0.62 ^a^	3.08 ± 0.09 ^b^	67.84 ± 0.99 ^ba^
Ap	15.79 ± 0.04 ^b^	13.21 ± 0.43 ^a^	3.39 ± 0.29 ^ba^	67.58 ± 0.82 ^ba^
Bc+Ap	26.65 ± 0.17 ^a^	14.42 ± 0.59 ^a^	3.41 ± 0.11 ^ba^	66.79 ± 0.57 ^b^

Values are means (*n* = 3) ± SD. Different letters represent a statistically significant difference among treatment (*p* < 0.05). Bc = *Bacillus coagulans* pasta; Ap = *A. platensis* pasta; Bc+Ap = pasta combining Bc+Ap.

**Table 6 foods-13-03381-t006:** Probiotic viability of *B. coagulans* in pasta formulations per gram.

Pasta Formulation	Uncooked	Cooked
Control	ND	ND
Bc	5.19 ± 0.01 ^b^	5.23 ± 0.02 ^b^
Ap	ND	ND
Bc+Ap	7 ± 0.01 ^a^	7.2 ± 0.17 ^a^

Values are means (*n* = 3) ± SD. Different letters represent a statistically significant difference among treatment (*p* < 0.05). Bc = *Bacillus coagulans* pasta; Ap = *A. platensis* pasta; Bc+Ap = pasta combining Bc+Ap.

**Table 7 foods-13-03381-t007:** Colorimetric analysis of pastas by CIELAB.

Pasta Sample	L*	a*	b*	Yellowness Index
Uncooked				
Raw Commercial control	80.07 ^c^	4.26 ^a^	33.17 ^a^	59.18
Control	91.46 ^a^	1.34 ^c^	12.65 ^c^	19.76
Ap	61.29 ^d^	7.99 ^d^	11.37 ^cd^	26.50
Bc	85.03 ^b^	3.18 ^b^	21.76 ^b^	36.56
Bc+Ap	49.95 ^e^	8.12 ^d^	10.15 ^d^	29.03
Cooked				
Cooked Commercial control	72.22 ^b^	7.45 ^a^	34.17 ^a^	67.59
Control	81.82 ^a^	3.35 ^b^	21.50 ^c^	37.54
Ap	54.56 ^c^	1.07 ^c^	20.95 ^c^	54.86
Bc	83.13 ^a^	3.63 ^b^	26.50 ^b^	45.54
Bc+Ap	47.70 ^d^	0.40 ^c^	18.76 ^c^	56.19

Values are means (*n* = 3) ± SD. Different letters represent a statistically significant difference among treatments (*p* < 0.05). Bc = *Bacillus coagulans* pasta; Ap = A. *platensis pasta*; Bc+Ap = pasta combining Bc+Ap. L* = lightness; a* = red/green coordinate; b* = yellow/blue coordinate.

## Data Availability

The original contributions presented in the study are included in the article, further inquiries can be directed to the corresponding authors.

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
