# Peer review of "Wheat Flour Pasta Combining Bacillus coagulans and Arthrospira platensis as a Novel Probiotic Food with Antioxidants"

_foods, 2024, doi:10.3390/foods13213381_

Round 1

Reviewer 1 Report

Comments and Suggestions for Authors

This study investigated the title " Wheat flour pasta combining Bacillus coagulans and Arthrospira platensis as a novel probiotic food with antioxidants" After revision, I think it is an interesting article.

- Significant Improvement: The study demonstrates that adding A. platensis and B. coagulans to pasta significantly enhances its nutritional value, antioxidant capacity, texture, and probiotic activity, offering potential health benefits and application prospects.

 Recommendations for Future Research or Study Limitations:

1. Consumer Acceptance Not Evaluated:

   - Recommendation: Conduct sensory evaluations and consumer surveys to assess the acceptance of pasta enriched with A. platensis and B. coagulans in terms of taste, color, and flavor. This would provide better insights into the market potential of the product.

2. Long-term Stability Not Assessed:

   - Recommendation: Perform long-term storage stability studies to evaluate changes in functional ingredients and probiotic viability under various storage conditions. This is crucial for determining the product’s shelf life.

3. Unclear Probiotic Dosage Effects:

   - Recommendation: Further investigate the effects of different dosages of B. coagulans on pasta’s functional properties and probiotic activity to determine the optimal amount for addition.

4. Probiotic Viability Research:

   - Recommendation: Conduct in-depth studies on the survival mechanisms of B. coagulans under different environmental conditions (e.g., stomach acid and intestinal environments) to better understand how probiotics function in the consumer's body.

5. Mechanism Exploration:

   - Recommendation: Explore the synergistic mechanisms between A. platensis and B. coagulans more thoroughly. This can be achieved through molecular biology techniques (e.g., gene expression analysis) to reveal how they jointly enhance antioxidant capacity and other functional characteristics.

6. Literature Citation:

   - Recommendation: Increase citations and discussion of existing literature, especially when comparing with other research results, to strengthen the credibility of the study’s findings.

Author Response

Comments 1:  Consumer Acceptance Not Evaluated:

   - Recommendation: Conduct sensory evaluations and consumer surveys to assess the acceptance of pasta enriched with A. platensis and B. coagulans in terms of taste, color, and flavor. This would provide better insights into the market potential of the product.

Response 1: Thank you for your feedback. We also consider that part critical of further research and will include it in the subsequent assays already planned.

Comments 2:  Long-term Stability Not Assessed:

   - Recommendation: Perform long-term storage stability studies to evaluate changes in functional ingredients and probiotic viability under various storage conditions. This is crucial for determining the product’s shelf life.

Response 2: We agree with this suggestion. It is crucial to evaluate the shelf life of functional foods to ensure that the potential health benefits provided by the bioactive compounds of the food are still in the same concentrations as those evaluated in the study. That study will be considered in further research.

Comments 3: Unclear Probiotic Dosage Effects:

   - Recommendation: Further investigate the effects of different dosages of B. coagulans on pasta’s functional properties and probiotic activity to determine the optimal amount for addition.

  Response 3: The minimal dosage of probiotic food (1x106) was considered a good initial point for this research. Still, we agree with exploring other higher concentrations and measuring their impact on the pasta formulation properties (textural and functional).

Comments 4: Probiotic Viability Research:

   - Recommendation: Conduct in-depth studies on the survival mechanisms of B. coagulans under different environmental conditions (e.g., stomach acid and intestinal environments) to better understand how probiotics function in the consumer's body.

Response 4: Thank you for your feedback. That point will be considered for further study; it is a great suggestion.

Comments 5: Mechanism Exploration:

   - Recommendation: Explore the synergistic mechanisms between A. platensis and B. coagulans more thoroughly. This can be achieved through molecular biology techniques (e.g., gene expression analysis) to reveal how they jointly enhance antioxidant capacity and other functional characteristics.

Response 5: Elucidating the synergism shown by probiotics and prebiotics could help us understand some of the intrinsically functional properties of pasta. Your feedback is valuable and greatly appreciated; it is an excellent point for further studies.

 Comments 6: Literature Citation:

   - Recommendation: Increase citations and discussion of existing literature, especially when comparing with other research results, to strengthen the credibility of the study’s findings.

Response 6: We considered just those studies comparing similar food commodities and methodologies for this research. Most studies used to potentiate this research have been selected based on the similarity of objectives and ingredients used. Still, other studies regarding food commodities added with A. platensis or B. coagulans were also considered valuable references. Many studies used to compare our findings reached the same conclusion regarding nutrimental analysis, pigment content, antioxidant capacity, and textural analysis. Although we have found some differences between the literature and our results, we considered those papers with similar results valuable. Unfortunately, scarce studies assess all the parameters evaluated in our research. Still, we have conducted an exhaustive search to collect all comparable studies, taking the parameters, procedures, and results as a basis. Thank you for your feedback. We took advantage of the review suggestions and added new studies to improve the introduction section. Furthermore, some document sections were improved (highlighted in yellow). Nonetheless, we hope this research serves as a guide for characterizing similar foods.

Reviewer 2 Report

Comments and Suggestions for Authors

1- page 1 line 22-24: what is meaning of Bc, ABTS &FRAP

2- page 1 line 24: pigment content ; instead of pigment content (chlorophyll a + b, and total carotenoids)

3- page 1 lines 27-32: the standard diviation for all results  ( The Bc+Ap pasta showed enhanced nutritional value with a significant increase in protein content (30.61%). After cooking, the pasta increased phenolic content (14.22% mg GAE/g), antioxidant capacity (55.59% µmol Trolox Equivalents/g and 10.88% µmol Fe+2/g) for ABTS and FRAP respectively, pigment content (6.72 and 1.17 mg/100g), for chlorophyll 

a+b and total carotenoids respectively, but relative positive impacts on colorimetric parameters in contrast to control (wheat flour pasta). Furthermore, Bc+Ap showed improved firmness (59%, measured in g), buffer capacity (87.80% μmol H+(g × ΔpH)−1), and good probiotic viability (7.2 ± 0.17 log CFU/g) after the cooking process.)should be add in abstract.

4- page 1 line36: key words;  Wheat flour pasta; functional; Arthrospira platensis; probiotic; antioxidant; novel foods. instead of  Pasta; functional; Arthrospira platensis; probiotic; antioxidant; novel foods.

5- page 6 line 273: Table 1 the meaning of Bc & Ap  should be written under table 

6- page 9 line 407 : Table 2 the meaning of Bc ,  Ap , TPC, ABTS & FRAP  should be written under table

7- page 17 line 773 : Table 7 the meaning of L*, a * & b*  should be written under table

8- page 20 lines 894-905: the conclusion  need improvement  to add the most relevant results in it 

9- page 20 line929: the references need to update 

Comments on the Quality of English Language

Minor editing of English language required.

Author Response

Comments 1:  page 1 line 22-24: what is meaning of Bc, ABTS &FRAP

Response 1: Thank you for your feedback. We have already attended to the suggestion in the new modified document. This change can be found page 1 line 19-25.

Comments 2: page 1 line 24: pigment content; instead of pigment content (chlorophyll a + b, and total carotenoids)

Response 2: Thank you for your feedback. We have already attended to the suggestion in the new modified document. This change can be found page 1 line 25.

Comments 3: page 1 lines 27-32: the standard diviation for all results (The Bc+Ap pasta showed enhanced nutritional value with a significant increase in protein content (30.61%). After cooking, the pasta increased phenolic content (14.22% mg GAE/g), antioxidant capacity (55.59% µmol Trolox Equivalents/g and 10.88% µmol Fe+2/g) for ABTS and FRAP respectively, pigment content (6.72 and 1.17 mg/100g), for chlorophyll a+b and total carotenoids respectively, but relative positive impacts on colorimetric parameters in contrast to control (wheat flour pasta). Furthermore, Bc+Ap showed improved firmness (59%, measured in g), buffer capacity (87.80% μmol H+(g × ΔpH)−1), and good probiotic viability (7.2 ± 0.17 log CFU/g) after the cooking process.)should be add in abstract.

Response 3: Thank you for your feedback. The use of percentages to compare the differences between the formulations could make it easy to understand the significance of the formulations assessed in this work, and the use of SD along with the percentages could predispose the reader to confusion. Your feedback is valuable, and we are considering that the SD should be placed in the tables, and the abstract should also be in terms of percentages to earn enough space to show all the research findings.

Comments 4: page 1 line36: key words; Wheat flour pasta; functional; Arthrospira platensis; probiotic; antioxidant; novel foods. instead of Pasta; functional; Arthrospira platensis; probiotic; antioxidant; novel foods.

Response 4: Thank you for pointing this out. We completely agree with your suggestion, and it was attended. This change can be found on page 1, line 36.

Comments 5: page 6 line 273: Table 1 the meaning of Bc & Ap should be written under table 

Response 5: Thank you for pointing this out. We completely agree with your suggestion, and it was attended. This change can be found on page 7, lines 307 and 308; Table 1.

Comments 6: page 9 line 407: Table 2 the meaning of Bc, Ap, TPC, ABTS & FRAP should be written under table

Response 6: Thank you for pointing this out. We completely agree with your suggestion, and it was attended. This change can be found on page 10, lines 457 – 459; Table 2.

Comments 7: page 17 line 773: Table 7 the meaning of L*, a * & b* should be written under table

Response 7: Thank you for pointing this out. We completely agree with your suggestion, and it was attended. This change can be found on page 18, lines 792 and 793: Table 7

Comments 8: page 20 lines 894-905: the conclusion need improvement  to add the most relevant results in it 

Response 8: Thank you for pointing this out. We consider that the conclusion includes all the most important/relevant research findings. However, we agree with your feedback, so we decided to add to the conclusion the percentages obtained for each parameter measured (analysis results). We hope that you agree with the new conclusion. We just wanted to make a clear and concise conclusion and suggest insights for further research on this topic. This change can be found on page 20, lines 890-908.

Comments 9: page 20 line929: the references need to update 

Response 9: Thank you for your feedback. Most of the references used for this work have been of recent publication; however, we are aware that some others are old, but those were considered valuable due to the lack of information in the actual literature, especially in lines 819-895 where unusual phenomena are being described and just pioneering papers can explain it. We exhaustively reviewed the literature to find actual references to explain these phenomena, but we all got pioneering papers describing similar phenomena. Nonetheless, this could be useful to highlight these ideas and be considered for further research. Another limitation during the collection of relevant studies was that this is the first study describing a formulation of this nature (combining A. platensis and B. coagulans) and assessing all the parameters shown here, which makes it more difficult to find similar results. Thank you for your time reviewing this document, we appreciate all your suggestions. This change can be found on pages 18 and 19, lines 819-895.

Reviewer 3 Report

Comments and Suggestions for Authors

This manuscript evaluates the effect of the combination of Arthrospira plantensis and Bacillus coagulans on uncooked and cooked pasta in terms of nutritional value, total phenolic content, antioxidant capacity, textural properties, pigment content, color, buffering capacity and probiotic vitality. It is well written and the experiments are carefully planned and conducted. Below are my detailed comments after reading the manuscript.

Line 138-140, It is stated that 5% of B. coagulans was added to the paste formulation and 1% is written in the abstract, which is correct?

Equations 1,2,3: please use word equation toolbox.

Table 1 What does the abbreviation hw mean - it is not given in the text either.

Line 300-301, Try to explain why the carbohydrate content decreased.

Line 463-465, Why the phycocyanin content was not analyzed?

Table 3. why the pigment content was not compared with the control pasta?

Line 717-740, Where are the results of textural analysis  for the raw pasta?

Section 3.2.6. Add to the title Effect of B. Coagulans addition on total phenolic content, antioxidant capacity and pigment content of pasta. The text is prescribed, extract the essence.

The manuscript is overwhelmed with numerous tables. Please transfer some tables to figures, if possible.

Author Response

Comments 1: Line 138-140, It is stated that 5% of B. coagulans was added to the paste formulation and 1% is written in the abstract, which is correct?

Response 1: Thank you for pointing this out. We completely agree with your suggestion, and it was attended. This change can be found on page 4, line 171.

Comments 2: Equations 1,2,3: please use word equation toolbox.

Response 2: Thank you for pointing this out. We already used the toolbox equations of Word to write the equations 1, 2, and 3. Please give us more details about this observation. This change can be found on page 5, lines 138-140.

Comments 3: Table 1 What does the abbreviation hw mean - it is not given in the text either.

Response 3: Thank you for pointing this out. We agree with your observation; it was attended to and modified to the correct meaning. We wanted to write dw (dry weight) instead of hw (that was a mistype). This change can be found on page 7, lines 305 and 307.

Comments 4: Line 300-301, Try to explain why the carbohydrate content decreased.

Response 4: Thank you for your comment. The carbohydrate amount was determined using a formula that considers all other macronutrients, which involves determining carbohydrates by difference. Since the protein parameter in the Bc+Ap pasta was higher, the carbohydrates obtained by difference were lower than those of the Ap or Bc pasta. However, all these analyses were carried out following AOAC protocols. This change can be found on page 7, lines 335 and 338.

Comments 4: Line 463-465, Why the phycocyanin content was not analyzed?

Response 4: That is an excellent suggestion. However, for practical reasons, this paper did not assess it. It is a useful insight to complement the potential health benefits of this formulation. We are considering that point in further investigations.

Comments 5: Table 3. why the pigment content was not compared with the control pasta?

Response 5: Thank you for your comment. It was compared to just those pasta with pigments. Since the formula (lines 242-246) used for determining the quantity of pigments requires that the food product contains chlorophylls and carotenoids, if one of these pigments is missing from the equation, it could give no exact value and, therefore, conduct an inadequate estimation of the content of pigments. To avoid that, just those formulations were considered.

Comments 6: Line 717-740, Where are the results of textural analysis for the raw pasta?

Response 6: Thank you for your comment. We decided to include only the results of the TPA analysis for cooked pasta because the tendency for reinforcement was more evident than for uncooked pasta. Uncooked pasta showed improvement, but there was no correlation between formulation and improvement in the TPA analysis. Regarding cooked pasta, a clear tendency of reinforcement was evidenced, making clear the effect of the different pasta ingredients (Ap and Bc) on TPA parameters. Furthermore, since the cooked TPA could give an idea of the form in which the food product will be prepared and consumed, it is more useful to provide a general idea of the TPA properties in the final format of the pasta.

Comments 7: Section 3.2.6. Add to the title Effect of B. Coagulans addition on total phenolic content, antioxidant capacity and pigment content of pasta. The text is prescribed, extract the essence.

Response 7: Thank you for pointing this out. The changes have been made in the new document and are highlighted in yellow. This change can be found on pages 18 and 19, lines 819 - 895.

Comments 8: The manuscript is overwhelmed with numerous tables. Please transfer some tables to figures, if possible.

Response 8: Thank you for pointing this out. We think that showing the results in tables with numerical values could make the reader more aware of the statistical analysis and make it easier to compare the results obtained for each formulation in the different assays.